# Person Re-Identification Method Based on Dual Descriptor Feature Enhancement

**DOI:** 10.3390/e25081154

**Published:** 2023-08-01

**Authors:** Ronghui Lin, Rong Wang, Wenjing Zhang, Ao Wu, Yang Sun, Yihan Bi

**Affiliations:** 1School of Information and Cyber Security, People’s Public Security University of China, Beijing 100038, China; nuaappsuclrh@126.com (R.L.); wenjingppsuc@126.com (W.Z.);; 2Key Laboratory of Security Prevention Technology and Risk Assessment of Ministry of Public Security, Beijing 100038, China

**Keywords:** person re-identification, dual network, face recognition, neural network

## Abstract

Person re-identification is a technology used to identify individuals across different cameras. Existing methods involve extracting features from an input image and using a single feature for matching. However, these features often provide a biased description of the person. To address this limitation, this paper introduces a new method called the Dual Descriptor Feature Enhancement (DDFE) network, which aims to emulate the multi-perspective observation abilities of humans. The DDFE network uses two independent sub-networks to extract descriptors from the same person image. These descriptors are subsequently combined to create a comprehensive multi-view representation, resulting in a significant improvement in recognition performance. To further enhance the discriminative capability of the DDFE network, a carefully designed training strategy is employed. Firstly, the CurricularFace loss is introduced to enhance the recognition accuracy of each sub-network. Secondly, the DropPath operation is incorporated to introduce randomness during sub-network training, promoting difference between the descriptors. Additionally, an Integration Training Module (ITM) is devised to enhance the discriminability of the integrated features. Extensive experiments are conducted on the Market1501 and MSMT17 datasets. On the Market1501 dataset, the DDFE network achieves an mAP of 91.6% and a Rank1 of 96.1%; on the MSMT17 dataset, the network achieves an mAP of 69.9% and a Rank1 of 87.5%. These outcomes outperform most SOTA methods, highlighting the significant advancement and effectiveness of the DDFE network.

## 1. Introduction

In the context of an increasingly information-driven society, intelligent surveillance facilities have assumed a critical role within social security systems. As an integral part of computer vision tasks, person re-identification holds indispensable significance within intelligent surveillance systems. Its primary objective lies in accurately identifying and matching person images obtained from different cameras [1], thereby facilitating real-time pedestrian tracking and comprehensive trajectory analysis [2]. Consequently, it serves as a powerful tool for efficiently analyzing intelligent surveillance data, contributing to the overall enhancement of security measures.

With the advancement of deep learning, traditional methods for person re-identification are gradually being replaced by deep learning approaches such as convolutional neural networks (CNN) and transformers. Currently, mainstream methods for deep pedestrian re-identification can be categorized into representation learning-based methods and metric learning-based methods, depending on their objectives and motivations. Representation learning-based methods [3,4,5] focus on the feature extraction process, while metric learning-based methods [6,7,8] concentrate on optimizing the metric space to obtain discriminative features with small inter-class distances and large intra-class distances. These methods adhere to a common paradigm: taking a pedestrian image as input, extracting features to obtain a descriptive feature descriptor that represents the image, and employing this descriptor for image matching. This pattern recognition paradigm closely resembles how humans observe objects. When presented with an unfamiliar person image, we initially form a general description in our minds and then proceed to identify the person based on that description. However, different observers may have distinct observations of the same image. For instance, some individuals emphasize body posture while others focus on facial features. Consequently, each observer acquires only partial descriptive information about the person image, as depicted in Figure 1a. Therefore, by integrating the descriptions provided by multiple observers of the same pedestrian image, as illustrated in Figure 1b, we can gather more comprehensive information and enhance the recognition of the unfamiliar pedestrian. Drawing upon insights and considering the influence of model size, this paper introduces a novel approach called the Dual Descriptor Feature Enhancement (DDFE) network. In this approach, two sub-networks act as separate observers, extracting features from the same person image from different viewpoints and generating two descriptors. These descriptors are subsequently integrated seamlessly to create a feature representation that captures the multi-view information for person image matching.

Assuming that we have obtained an integrated description of the same person image by two observers, the key way to improve the final recognition effect is to discern the ways to make this integrated description more comprehensive and detailed. Continuing from the perspective of human recognition patterns, it is evident that the first step is to ensure that each observer provides detailed and accurate descriptions of the observed subject, as depicted by enlarging the areas of the two color ellipses in Figure 2a,b. At the same time, efforts should be made to maximize the differences between the feature descriptions, making them complementary to each other. This enhances the overall observations in the final integrated description, as shown in Figure 2b,c. Building on these insights, this paper carefully designs the training strategy for the DDFE network. Firstly, it introduces the CurricularFace loss [9], a technique borrowed from the field of face recognition, to effectively guide each sub-network in obtaining superior representations of the input pedestrian images. Secondly, it incorporates the DropPath [10] operation, which introduces randomness into the training process, thereby preventing the two sub-networks from generating identical features and promoting greater diversity in the feature descriptors. Finally, an Integration Training Module (ITM) is proposed to seamlessly merge the feature descriptors from the two sub-networks. This module aims to enhance the discriminability of the integrated features while ensuring consistency between the objectives pursued during the model training stage and the inference stage.

In summary, this paper makes the following key contributions.
Proposes a Dual Descriptor Feature Enhancement (DDFE) network that extracts and integrates features from two perspectives of person images, resulting in more discriminative features and improved recognition accuracy of the model;Designs targeted training strategies for the Dual Descriptor Feature Enhancement network, including the incorporation of CurricularFace loss, DropPath operation, and the Integration Training Module;Tests extensively on datasets Market1501 and MSMT17, demonstrating state-of-the-art performance in person re-identification.

## 2. Related Work

### 2.1. Person Re-Identification

The current mainstream methods for person re-identification can be classified into two categories: representation learning-based methods and metric learning-based methods.

Representation learning-based methods aim to improve the feature extraction process of deep models by generating highly discriminative and semantically meaningful features for more accurate pedestrian image matching. Among them, based on whether the models utilize local images or local features during the feature extraction process, representation learning-based methods can be further categorized into global feature-based methods and local feature-based methods. Global feature-based methods [4,11,12] take the entire pedestrian image as the model input and directly extract features for subsequent recognition. These methods have clear ideas and simple model flows, but improving the discriminative power of the model’s output features under global input becomes a key point for improving the accuracy of such methods. Local feature-based methods focus on the local details in the image, commonly achieved through predefined image or feature map segmentation [13,14,15], attention mechanisms [16,17], or the incorporation of human pose estimation [18] or body parsing [19]. Metric learning-based methods refer to the related approaches that optimize the feature metric space using different loss functions during model training. In the mainstream training paradigm of pedestrian re-identification methods, cross-entropy loss is commonly used as the classification loss, and triplet loss is employed as the metric loss. Currently, there have been much improvement in this paradigm with various metric learning methods [6,7,8,20].

### 2.2. Face Recognition Loss Function

The cross-entropy loss function with Margin (angle penalty) is commonly used in face recognition model training to enhance intra-class compactness and inter-class separability of the output features. SphereFace loss [21] introduces the transformation of the face recognition feature space into angular cosine space and applies Margin penalty to the angle between samples and their class centers, aiming for improved inter-class and intra-class distances. CosFace loss [22] builds upon SphereFace by scaling unit-normalized feature vectors onto a hypersphere with radius s and directly penalizing the cosine value of the angle between samples and class centers, addressing SphereFace’s convergence issue. ArcFace loss [23] further enhances the discriminative power by moving the penalty term inside the cosine function, directly penalizing the angle between samples and class centers. This modification improves training stability and feature discriminability. MV-Arc-Softmax loss [24] considers misclassified samples as hard samples, assigning them higher weights during model training to guide effective discriminative feature learning. CurricularFace loss [9] also treats misclassified samples as hard samples but incorporates curriculum learning into the face recognition loss function. It assigns smaller weights to the hard samples in the early training stages, gradually increasing their weights as the model trains. This mimics the human learning pattern of starting with easier samples and progressing to harder ones, resulting in more discriminative features. This paper takes inspiration from face recognition loss functions and applies them to person re-identification network training, improving recognition accuracy.

## 3. Methodology

In this section, the Dual Descriptor Feature Enhancement (DDFE) network proposed in this paper is introduced in Section 3.1, followed by a detailed explanation of the targeted training strategy for the DDFE network in Section 3.2.

### 3.1. Dual Descriptor Feature Enhancement (DDFE) Network

The overall architecture of the proposed DDFE network is depicted in Figure 3. It comprises two sub-networks with identical structures, which generate distinct descriptors for a given pedestrian image. In detail, the specific pipeline is as follows: First, a person image is inputted and the ConvNeXt V2 Tiny [25] feature extractors in both sub-networks perform feature extraction on the input image, resulting in feature maps. Second, the feature maps are passed through the generalized-mean pooling (GeM pooling) [26] layer with shared weights to obtain the feature vectors $f_{t}^{1}$ and $f_{t}^{2}$. Then, they are further processed via the batch normalization (BN) layer with shared parameters to yield the descriptors $f_{i}^{1}$ and $f_{i}^{2}$. Third, the descriptors $f_{i}^{1}$ and $f_{i}^{2}$ are element-wise added together to obtain the final integrated feature f. Finally, the Euclidean distance between the integrated features f of different person images is computed for image matching. It is worth noting that sharing the weight parameters in the GeM pooling layer and the BN layer aims to align the features extracted via the two sub-networks into a common distribution, ensuring that the differences observed in the descriptors originate solely from the variations in the “observers” performing feature extraction.

Among them, the ConvNeXt V2 Tiny model predominantly comprises 18 Block structures, as illustrated in Figure 4. These 18 Blocks are stacked in four layers with a distribution of [3, 3, 9, 3]. The Block also incorporates Global Response Normalization (GRN), which fosters competition among the channels for enhanced feature representation, as demonstrated in Equation (1).
(1)X¯i=γ×Xi×NXi+β+XiNXi=Xi∑j=1,…,CXj
where X∈ℝC×H×W represents the feature map before Global Response Normalization (GRN), with a total of C channels. Xi∈ℝH×W denotes the *i*-th channel feature map of X. Xi represents the L2 norm of Xi. γ and β are learnable parameters. X¯i represents the *i*-th channel feature map after GRN.

### 3.2. Training of the DDFE Network

In this subsection, a general overview of the whole training stage of the DDFE network is given in Section 3.2.1, then the main parts of the training strategy, including DropPath, Integration Training Module (ITM) and CurricularFace loss, are described in detail in Section 3.2.2, Section 3.2.3 and Section 3.2.4, and finally in Section 3.2.5 the total loss for the training is introduced.

#### 3.2.1. Overview

To enhance the representational capacity of the DDFE network, this paper focuses on three aspects during the training stage. Firstly, it aims to enhance the recognition accuracy of each sub-network. Secondly, it aims to increase the differences between the output features of the sub-networks to enhance the information content in the integrated feature. Lastly, it aims to improve the discriminability of the integrated features of the two sub-networks while aligning the training objective with the inference objective.

Based on the above, the specific pipeline of the training stage of the DDFE network is shown in Figure 5. Firstly, DropPath [10] operations are introduced in the ConvNeXt V2 Tiny model to introduce randomness and diversity between the sub-networks. Next, the feature vectors $f_{t}^{1}$ and $f_{t}^{2}$ are fed into the Integration Training Module, where the integration loss $Loss_{\mathrm{Integration}}$ is computed to enhance the discriminability of the integrated features. Lastly, the Weighted Regularization Triplet (WRT) loss [2] is calculated for the feature vectors $f_{t}^{1}$ and $f_{t}^{2}$ to optimize the inter-class and intra-class distances in the Euclidean space. Moreover, the CurricularFace loss is computed for the descriptors $f_{i}^{1}$ and $f_{i}^{2}$ obtained after the BN layer to optimize the feature distances in the cosine space.

#### 3.2.2. DropPath

During the model training stage, DropPath [10] is introduced into ConvNeXt V2 Tiny, randomly deactivating each Block in ConvNeXt V2 Tiny. However, during the model inference stage, DropPath operations are not applied.

As shown in Figure 6, the specific implementation of DropPath is as follows: Firstly, we set an initial DropPath probability for the model as P. Then, the DropPath probability for each Block is calculated based on P. Specifically, the DropPath probability for the *i*-th Block, denoted as $P_{i}$, is calculated as $P_{i}=P/i$. Lastly, for each Block, the structures within it are either discarded with a probability of $P_{i}$ or retained with a probability of $1-P_{i}$. This means that when an input enters a Block, it has a probability of $P_{i}$ to be directly outputted without any computation. With a probability of $1-P_{i}$, the input passes through all the structures within the Block, undergoes a rescale operation (divided by $1-P_{i}$), and is then outputted after being connected with the input through a residual connection.

The use of DropPath serves two main purposes. Firstly, it acts as a regularization technique for each sub-network, preventing overfitting during the training process. Secondly, by randomly deactivating the Blocks in both sub-networks, it significantly increases the differences between them. This ensures that the two sub-networks do not generate identical features, leading to a more comprehensive description of the input image in the final integrated feature.

#### 3.2.3. Integration Training Module (ITM)

As the inference stage of the model involves integrating the output feature descriptors from both sub-networks to match person images, it is crucial to also integrate the output features of the sub-networks during the model training stage. This ensures consistency between the training and inference stages, thereby enhancing the discriminability of the final integrated feature during inference. To address this, this paper introduces an Integration Training Module (ITM), as depicted in Figure 7.

The ITM follows a step-by-step procedure: First, the feature vector $f_{t}^{1}$ from sub-network 1 and the feature vector $f_{t}^{2}$ from sub-network 2 are concatenated. Then, they pass through a fully connected layer to integrate the two feature vectors, resulting in the integrated feature $f_{\mathrm{Integration}}$. The Weighted Regularization Triplet (WRT) loss [2] is computed to optimize the relative distances of the integrated features in the Euclidean space. Finally, the integrated feature $f_{\mathrm{Integration}}$ undergoes through a BN layer (with weight parameters shared with the BN layers in the sub-networks) and another fully connected layer to obtain the Logits vector. The SoftMax function is applied to the Logits vector to calculate the cross-entropy loss.

The total loss of the Integration Training Module is the sum of the WRT loss and the cross-entropy loss, as shown in Equation (2), where the WRT loss is shown in Equation (3).
(2)$$Loss_{\mathrm{Integration}}=Loss_{\mathrm{WRT}}+\lambda_{1}Loss_{\mathrm{Cross}-\mathrm{Entropy}}$$
where $Loss_{\mathrm{WRT}}$ is the WRT loss, $Loss_{\mathrm{Cross}-\mathrm{Entropy}}$ is the cross-entropy loss, and $\lambda_{1}$ is the weight hyper-parameter.
(3)$$\begin{array}{c} Loss_{wrt}=\frac{1}{N}\sum_{i=1}^{N}\log\left(1+\exp\left(\sum_{ij}w_{ij}^{p}d_{ij}^{p}-\sum_{ik}w_{ik}^{n}d_{ik}^{n}\right)\right)\\ w_{ij}^{p}=\frac{\exp\left(d_{ij}^{p}\right)}{\sum_{d_{ij}^{p}\in P_{i}}\exp\left(d_{ij}^{p}\right)},w_{ik}^{n}=\frac{\exp\left(-d_{ik}^{n}\right)}{\sum_{d_{ik}^{n}\in N_{i}}\exp\left(-d_{ik}^{n}\right)} \end{array}$$
where i is any training sample; j denotes any positive sample of i; k denotes any negative sample of sample i; $P_{i}$ denotes the set of positive samples of i; $N_{i}$ denotes the set of negative samples of i; $d_{ij}^{p}$ denotes the Euclidean distance between i and any positive sample; $d_{ik}^{n}$ denotes the Euclidean distance between i and any negative sample; N denotes the number of samples within a training batch; $\sum_{ij}w_{ij}^{p}d_{ij}^{p}$ denotes the weighted sum of all positive sample distances of i; and $\sum_{ik}w_{ik}^{n}d_{ik}^{n}$ denotes the weighted sum of all negative sample distances of i.

#### 3.2.4. CurricularFace Loss

CurricularFace loss [9] is a metric loss function proposed in the field of face recognition. This loss function mimics the learning process of humans, where easier samples are given higher weights during the initial stages of training, while harder samples are given higher weights in the later stages of model training. This approach allows the model to achieve better local optima and produce more discriminative features. The formula for CurricularFace loss is shown as Equation (4).
(4)$$\begin{array}{c} Loss_{\mathrm{CurrilarFace}}=-\log\frac{e^{s\cos(\theta_{y_{i}}+m)}}{e^{s\cos(\theta_{y_{i}}+m)}+\sum_{j=1,j\neq y_{i}}^{n}e^{sN(t^{\left(k\right)},\cos\theta_{j})}}\\ N(t,\cos\theta_{j})=\left\{\begin{array}{cc} \cos\theta_{j}, & \cos(\theta_{y_{i}}+m)-\cos\theta_{j}\geq 0\\ \cos\theta_{j}(t+\cos\theta_{j}), & \cos(\theta_{y_{i}}+m)-\cos\theta_{j}<0 \end{array}\right.\\ t^{\left(k\right)}=\alpha r^{\left(k\right)}+(1-\alpha )t^{\left(k-1\right)} \end{array}$$
where s is the value of the rescaled modal length of the feature vector in cosine space (hyper-spherical radius); m is the angle penalty term; $\theta_{y_{i}}$ is the angle between the feature vector in cosine space and its corresponding $y_{i}$ category center; $\theta_{j}$ is the angle between the feature vector and the other category centers j; the relationship between $\cos(\theta_{y_{i}}+m)$ and $\cos\theta_{j}$ is used to define easy and hard samples, respectively; n is the number of category centers; $r^{\left(k\right)}$ is the average cosine similarity between the feature vectors of the current training batch (batch k) and their corresponding class centers, which increases as the model is trained; $t^{\left(k\right)}$ is the exponential moving average of $r^{\left(k\right)}$; and α is the momentum parameter of 0.99. At the beginning of training, $t^{\left(0\right)}=0$ is initialized as 0, as the model trains, t increases and gradually approaches 1, which is used to characterize the training process of the model (early and late).

#### 3.2.5. Total Loss

The total loss during the model training stage, denoted as $Loss_{\mathrm{Total}}$, is composed of three components: the loss of sub-network 1 ($Loss_{\mathrm{Network}1}$), the loss of sub-network 2 ($Loss_{\mathrm{Network}2}$), and the loss of the ITM ($Loss_{\mathrm{Integration}}$), as shown in Equation (5). Each sub-network loss is further composed of the WRT loss and the CurricularFace loss, as illustrated in Equation (6).
(5)$$Loss_{\mathrm{Total}}=Loss_{\mathrm{Network}1}+Loss_{\mathrm{Network}2}+Loss_{\mathrm{Integration}}$$
(6)$$Loss_{\mathrm{Network}1\;\mathrm{or}\;2}=Loss_{\mathrm{WRT}}+\lambda_{2}Loss_{\mathrm{CurrilarFace}}$$
where $\lambda_{2}$ is the weight hyper-parameter for CurricularFace loss.

## 4. Experiments and Analysis

### 4.1. Datasets and Evaluation Metric

The proposed method was experimented and evaluated on two publicly available person re-identification datasets: Market1501 [27] and MSMT17 [28]. The specific details of these two datasets are presented in Table 1.

The Market1501 dataset includes 32,668 images of 1501 individuals from a shopping center, captured via six cameras. It has a training set of 12,936 images of 751 persons and a testing set of 19,732 images of 750 persons.

The MSMT17 dataset consists of 126,441 images of 4101 pedestrians captured via 15 cameras, 12 outdoor and 3 indoor cameras. The training set has 32,621 images of 1041 persons and the testing set includes 82,161 images of 3010 persons.

The evaluation metrics used are Rank-k, mAP (mean average precision), and mINP (mean inverse negative penalty) [2]. The Rank-k metric assesses a model’s capability to accurately identify correct samples within the top k positions of the Rank List. In this paper, Rank1 is employed as the evaluation metric, specifically measuring the model’s accuracy in correctly identifying the samples at the first position in the Rank List. The mAP metric measures the recognition accuracy of the model, with higher mAP values indicating superior model performance. The mINP is a metric that quantifies the cost associated with locating the most challenging matching correct sample. In essence, it signifies that the deeper the last correct sample is ranked within the Rank List, the higher the cost of manual scrutiny and intervention. A lower mINP value indicates inferior model performance, as demonstrated in Equation (7).
(7)$$mINP=\frac{1}{n}\sum_{i=1}^{n}\frac{\left|G_{i}\right|}{R_{i}^{hard}}$$
where $\left|G_{i}\right|$ denotes the number of all correct samples of Query i, $R_{i}^{hard}$ denotes the position of the last correct sample of Query i in the Rank List, and n denotes the number of Query in the test set.

### 4.2. Experimental Settings

The experiments were conducted using the PyTorch deep learning framework version 1.4.0. The training was performed on a single machine with dual GPUs, specifically a Nvidia Titan V and a Nvidia GeForce RTX 2080. During model training, the input person images were resized to 256 × 128, and were subjected to random cropping, random erasing (with a probability of 0.5), and random horizontal flipping. No data augmentation was applied during the inference stage of the model. During training on the Market1501, the initial DropPath probability P was 0.1, weight hyper-parameter $\lambda_{1}$ in the $Loss_{\mathrm{Integration}}$ was 0.05, m and s in the CurricularFace loss were 0.4 and 30, respectively, weight hyper-parameter of the CurricularFace loss $\lambda_{2}$ was 0.05; during training on the MSMT17 dataset, the initial DropPath probability P was 0.2, $\lambda_{1}$ was 1, m and s were 0.3 and 45, respectively, and $\lambda_{2}$ was 0.01. The ConvNeXt V2 Tiny model is initialized with pre-trained weights from ImageNet, and the stride of the final downsampling convolutional layer in it is adjusted to 1 to enhance the resolution of the resulting feature map. The number of epochs for model training was 120, and the size of each training batch was 64, containing 16 persons with 4 images each. The initial learning rate was set to 0.0105 using the Adam optimizer, and it was fine-tuned with the WarmUp strategy. As is shown in Equation (8), the WarmUp strategy gradually increases the learning rate during the early stages of training and gradually decreases it as training progresses. This approach aims to enhance the model’s stability and convergence throughout the training process.
(8)$$lr\left(epoch\right)=\left\{\begin{array}{cc} 0.0105\times \frac{epoch+1}{10}, & epoch<10\\ 0.0105, & 10\leq epoch<40\\ 0.00105, & 40\leq epoch<70\\ 0.000105, & epoch\geq 70 \end{array}\right.$$

### 4.3. Comparison with Existing Methods

On the Market1501 and MSMT17 datasets, the Dual Descriptor Feature Enhancement (DDFE) network was compared with state-of-the-art methods including CF-ReID [29], PGANet-152 [30], CAL [17], TransReID [5], TMP [15], DCAL [31], LTReID [32], UniHCP [33], CLIP-ReID [34], and DC-Former [35]. The comparison results are shown in Table 2.

Based on the analysis of “Backbone during the inference stage (Params)” and “Training cost” in Table 2, it is evident that the proposed DDFE network, despite employing two sub-networks simultaneously, falls between ResNet101 [36] and ViT-Base [37] in terms of model size and parameter count. Additionally, the training cost of the DDFE network is also positioned within the middle range compared to the other methods.

In terms of recognition performance, the DDFE network achieves outstanding results. On the Market1501 dataset, it surpasses all other methods with an mAP of 91.6% and a Rank1 accuracy of 96.1%. Similarly, on the MSMT17 dataset, it outperforms all the methods, except CLIP-ReID, achieving an mAP of 69.9% and a Rank1 accuracy of 87.5%.

These comparative results indicate that the proposed DDFE network effectively enhances the model’s recognition performance while maintaining a moderate parameter count and training cost. This provides strong evidence for the effectiveness and advancement of the proposed method.

### 4.4. Ablation Study

#### 4.4.1. Performance Comparison: Sub-Network Descriptors vs. Integrated Feature

During the inference stage of the DDFE network, the recognition performance of the descriptor $f_{i}^{1}$ from sub-network 1, the descriptor $f_{i}^{2}$ from sub-network 2, and the integrated feature f were tested on the Market1501 dataset. This analysis aims to validate the effectiveness of integrating the descriptors from the two sub-networks for image matching, as shown in Table 3.

From Table 3, it is evident that integrating the descriptors from both sub-networks significantly enhances the network’s mAP and mINP values, while the Rank1 value remains unchanged. The lack of improvement in the Rank1 value can be attributed to its representation of the model’s performance on simple samples. For the Market1501 dataset, both the sub-network features and integrated features effectively capture information from simple person images, resulting in a performance bottleneck for Rank1. On the other hand, the mAP value reflects the model’s retrieval capability for both simple and challenging samples, while the mINP value signifies its ability to handle the most difficult samples. Utilizing the integrated features for person image matching provides a more comprehensive description from various perspectives, particularly benefiting challenging samples and significantly improving the mAP and mINP values of the network.

#### 4.4.2. Component Ablation Experiments of the Training Strategy

In order to assess the impact of each component of the training strategy designed for the DDFE network on the network’s recognition performance during the inference stage, a series of ablation experiments were conducted and trained and tested on the Market1501 dataset. The results are summarized in Table 4, where “√” indicates the inclusion of DropPath, Integration Training Module (ITM), or CurricularFace loss during model training.

In the Table 4, Experiment 1 involves constructing a person re-identification model using two sub-networks. During the training stage, DropPath is not utilized, and the CurricularFace loss is replaced with cross-entropy loss. In the inference stage, the descriptors generated via the two sub-networks are summed and employed for person image matching. Experiments 2 to 4 expand upon Experiment 1 by gradually introducing additional training components, such as DropPath, Integration Training Module (ITM), and the CurricularFace loss. Notably, Experiment 4 represents the DDFE network proposed in this paper.

Based on the results presented in Table 4, it is evident that each training component significantly enhances the model’s recognition performance during the inference stage. The utilization of DropPath, the ITM, and CurricularFace leads to the highest recognition performance, with an mAP of 91.6%, Rank1 of 96.1%, and mINP of 75.1%. These outcomes from the ablation experiments effectively showcase the effectiveness of each component within the training strategy of the proposed DDFE network.

#### 4.4.3. Generalizability Analysis

Various convolutional backbone networks, including ResNet-50 [36], MogaNet-T [38], and HorNet-T (GF) [39], as feature extractors, are used to assess the generalization of the proposed dual descriptor enhancement paradigm and model training strategy. The experimental results are presented in Table 5. In these experiments, the “Single Network” configuration utilized a single backbone network as the feature extractor for both model training and inference. DropPath operation was excluded during the training stage and the CurricularFace loss was replaced with cross-entropy loss. The “Dual Network” involves utilizing the feature extraction network to construct the proposed DDFE network, incorporating DropPath, CurricularFace loss, and the ITM during the training stage. All experiments were conducted without any fine-tuning. From the results in Table 5, it is evident that the proposed dual descriptor enhancement paradigm and model training strategy significantly enhance the recognition performance of person re-identification models across different backbone networks, thereby demonstrating the generalizability of the proposed approach.

#### 4.4.4. Parameter Analysis

For all hyper-parameters in the DDFE network, including the initial DropPath probability P, the weight hyper-parameter $\lambda_{1}$ in the $Loss_{\mathrm{Integration}}$, the m and s in the CurricularFace loss, as well as the weight of the CurricularFace loss $\lambda_{2}$, an analysis was conducted. The experimental details and results are presented in Figure 8 and Figure 9. From Figure 8, it can be observed that for the Market1501 dataset, the optimal hyper-parameters are m = 0.4, s = 30, $\lambda_{1}$ = 0.05, $\lambda_{2}$ = 0.05, and P = 0.1. Similarly, from Figure 9, it can be seen that for the MSMT17 dataset, the optimal parameters are m = 0.3, s = 45, $\lambda_{1}$ = 1, $\lambda_{2}$= 0.01, and P = 0.2.

### 4.5. Visualization Analysis

#### 4.5.1. Attention Heatmap Analysis of Each Sub-Network

To gain a comprehensive understanding of how sub-network 1 and sub-network 2 focus on person images in the DDFE network, we employed Grad-Cam [40] to visualize the attention heatmaps of the last convolutional layer’s feature maps for each sub-network. The visualizations are depicted in Figure 10. From the heatmaps in Figure 10, it is evident that the two sub-networks exhibit distinct areas of focus for the same input image. By integrating the output features of both sub-networks, a more comprehensive description can be obtained from two different attention perspectives, leading to an improved recognition accuracy for the person re-identification model.

#### 4.5.2. Visualization and Analysis of the Training Strategy

To better understand the impact of the designed training strategy on the recognition performance of the DDFE network, we randomly selected a person image from the test set of the Market1501 dataset as the query. The recognition results of each experimental model in the “Section 4.4.2 Ablation Experiments for Training Strategy” are displayed in Figure 11. The vertical numbers on the left correspond to the experimental numbers in Section 4.4.2. The “Query” represents the person image to be matched, and the images to the right of each query are the recognition results. The numbers above the recognition results indicate the similarity level between the recognition image and the query image, with smaller numbers indicating higher similarity (smaller Euclidean distance). The green box indicates correct recognition, while the red box indicates incorrect recognition.

Based on the findings from Experiment 1 in Figure 11, it is evident that the omission of DropPath, ITM, and CurricularFace loss leads to an incorrect recognition at the second position in the Rank List. However, with the inclusion of DropPath, the erroneous result now appears at the third position. Even when both DropPath and ITM are incorporated, the recognition performance still exhibits an incorrect recognition at the fourth position of the Rank List. Notably, the model achieves optimal training and recognition outcomes when all the proposed training strategies in this study are implemented, as demonstrated in Experiment 4 in Figure 11. Interestingly, the occurrence of incorrectly recognized images begins to manifest at the sixth position in the Rank List. This visualization effectively demonstrates the effectiveness of the proposed training strategy in this paper.

#### 4.5.3. Results Output

In this study, a random selection of five person images from the Market1501 dataset were used as query images. The DDFE network was utilized to extract feature descriptors for these query images. Concurrently, integrated features were also extracted for all person images in the gallery. Subsequently, the Euclidean distance between the integrated feature of each query image and the integrated feature of all gallery images was calculated. Finally, the top 10 images with the smallest Euclidean distances in the gallery were outputted and presented, as shown in Figure 12.

From the results in Figure 12, it can be observed that for simple frontal images of persons (Query1~3), the DDFE network successfully extracts rich information for feature description and recognition. For more challenging samples, such as images of persons from the back (Query4 and Query5), the DDFE network is also able to distinguish and recognize them to some extent among all the visually similar images.

## 5. Conclusions

This paper introduces a novel person re-identification method called Dual Descriptor Feature Enhancement (DDFE) network, which employs two sub-networks to extract features from the same person image. Compared to the traditional single-network approaches, this method provides better representation for person images. The incorporation of CurricularFace loss, DropPath operation, and Integration Training Module (ITM) during the model training phase substantially enhances the recognition performance of the DDFE network. Extensive experiments conducted on the Market1501 and MSMT17 datasets demonstrate the effectiveness, state-of-the-art performance, and generalization capability of the proposed method.

## Figures and Tables

**Figure 1 entropy-25-01154-f001:**
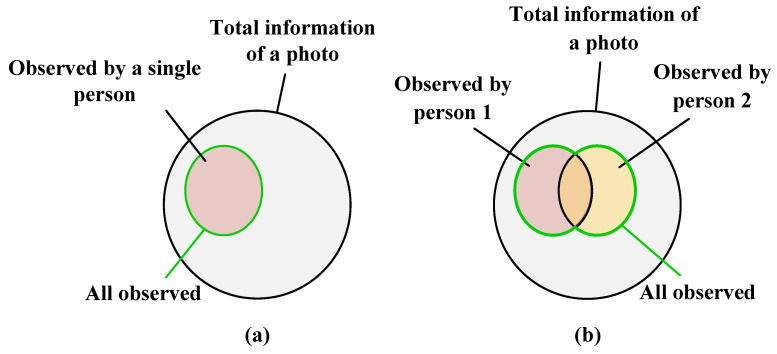
One observer and multiple observers. (**a**) One observer watches a picture. (**b**) multiple observers watch a photo.

**Figure 2 entropy-25-01154-f002:**
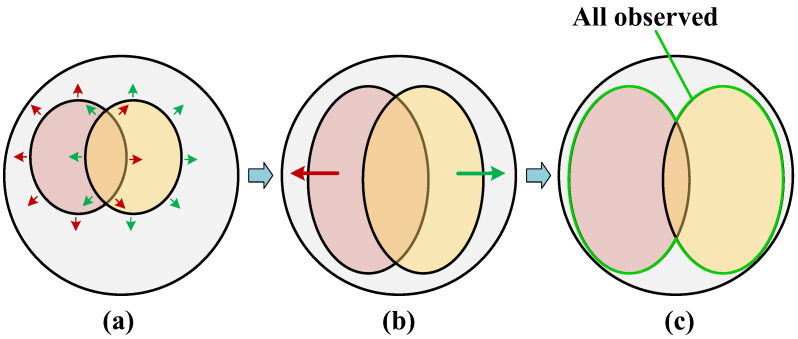
Illustration of how to increase the total observations by multiple observers. (**a**) Enhance the information observed by each observer. (**b**) Enhance the variability of information observed by the two observers. (**c**) The combined observations of the two observers after (**a**,**b**).

**Figure 3 entropy-25-01154-f003:**
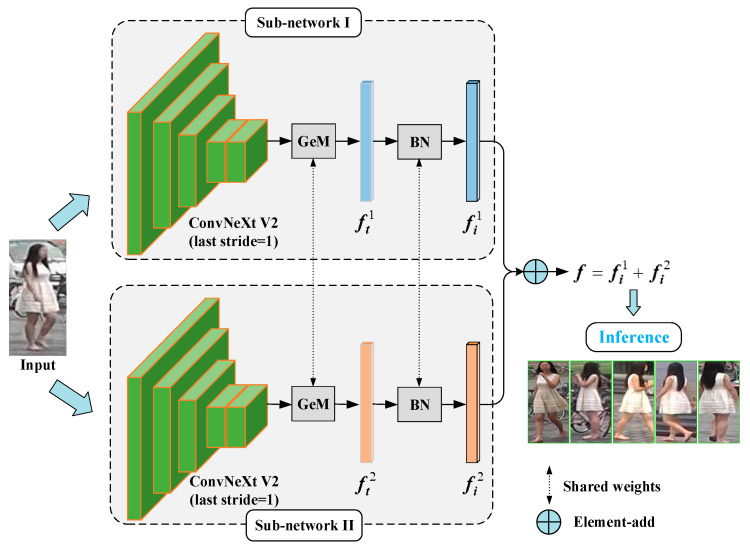
The Dual Descriptor Feature Enhancement (DDFE) network pipeline.

**Figure 4 entropy-25-01154-f004:**
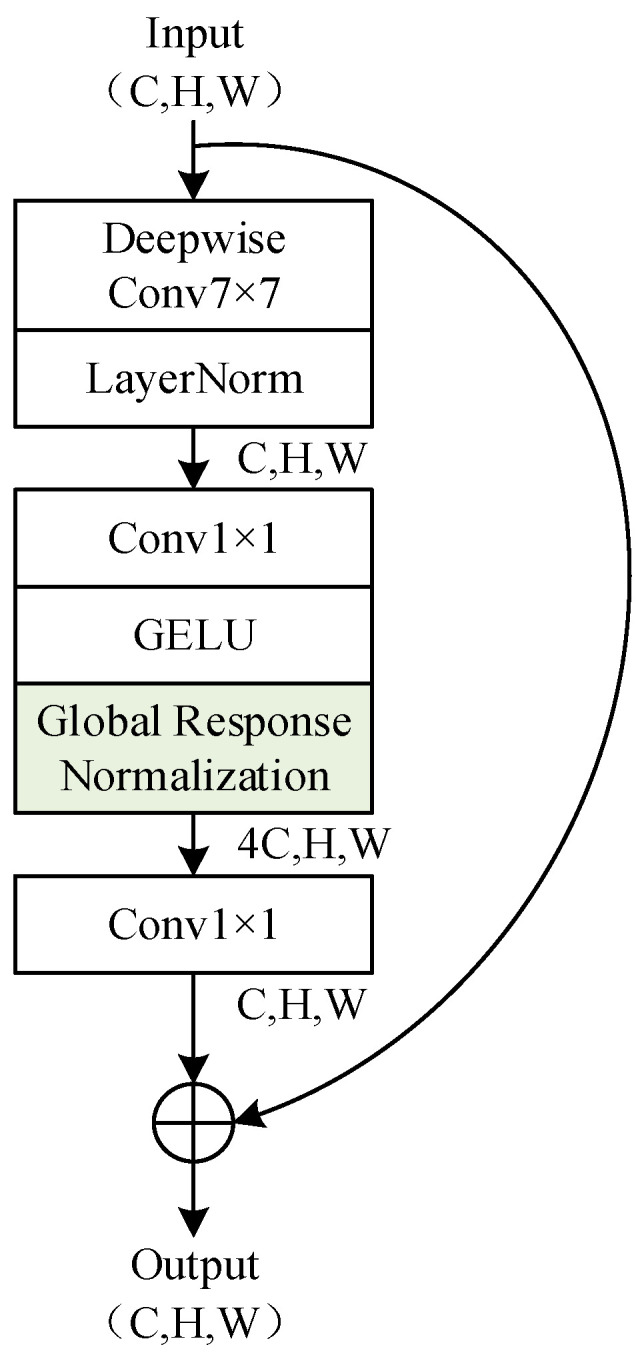
Block structure of the ConvNeXt V2 Tiny.

**Figure 5 entropy-25-01154-f005:**
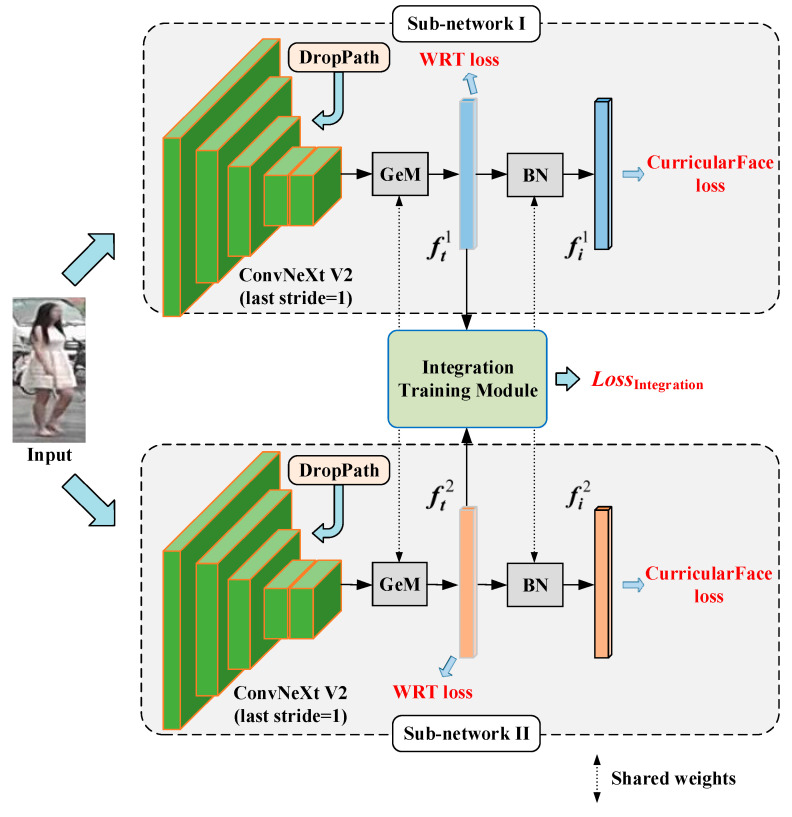
Training strategy of the DDFE network.

**Figure 6 entropy-25-01154-f006:**
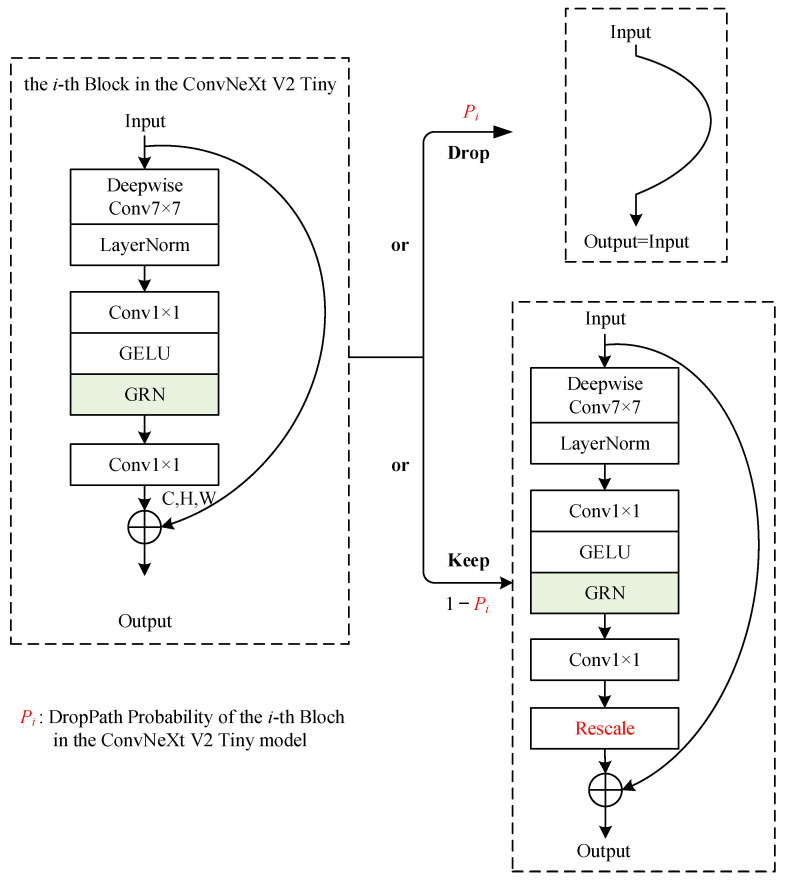
DropPath of the *i*-th Block in the ConvNeXt Tiny.

**Figure 7 entropy-25-01154-f007:**
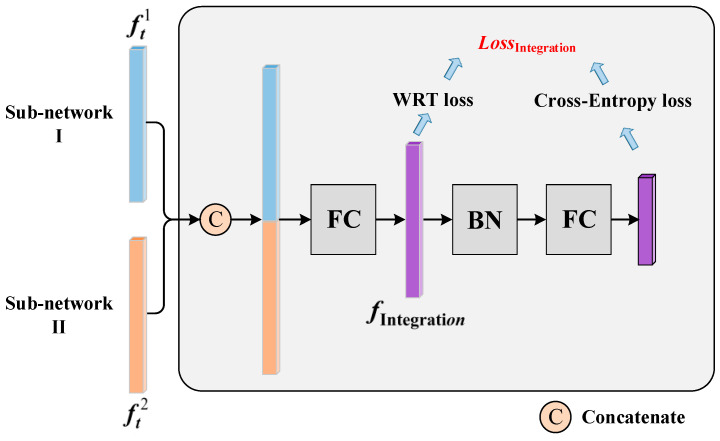
Integration Training Module (ITM).

**Figure 8 entropy-25-01154-f008:**
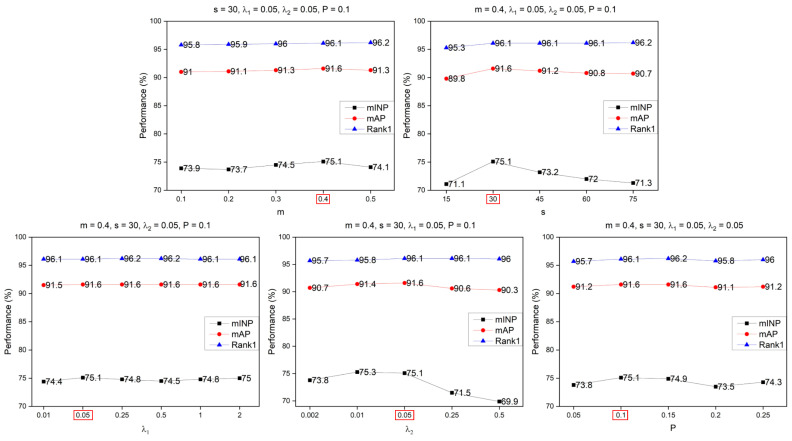
Parameter analysis in the Market1501. The red box on the horizontal axis of each subplot highlights the chosen optimal parameters.

**Figure 9 entropy-25-01154-f009:**
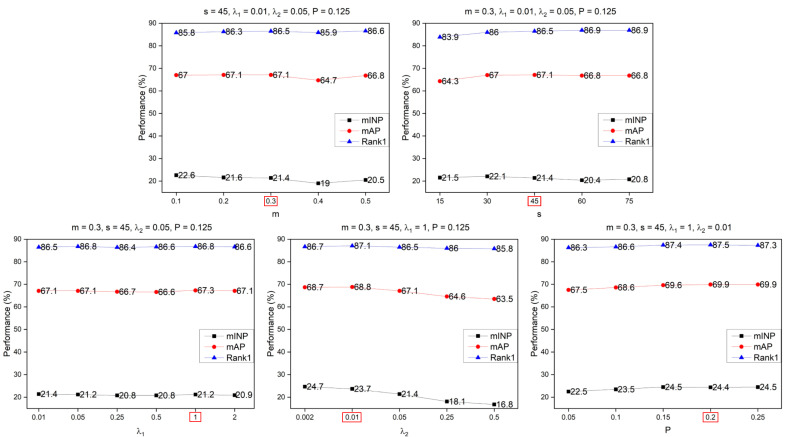
Parameter analysis in the MSMT17. The red box on the horizontal axis of each subplot highlights the chosen optimal parameters.

**Figure 10 entropy-25-01154-f010:**
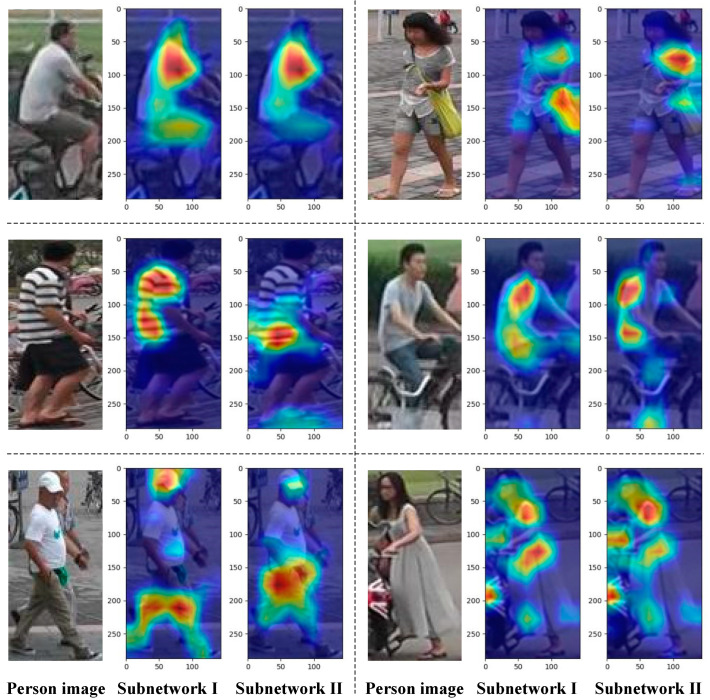
Analysis of attention heatmaps for each sub-network. Each set of three images includes the input person image, the attention heatmap for sub-network 1, and the attention heatmap for sub-network 2, arranged from left to right.

**Figure 11 entropy-25-01154-f011:**
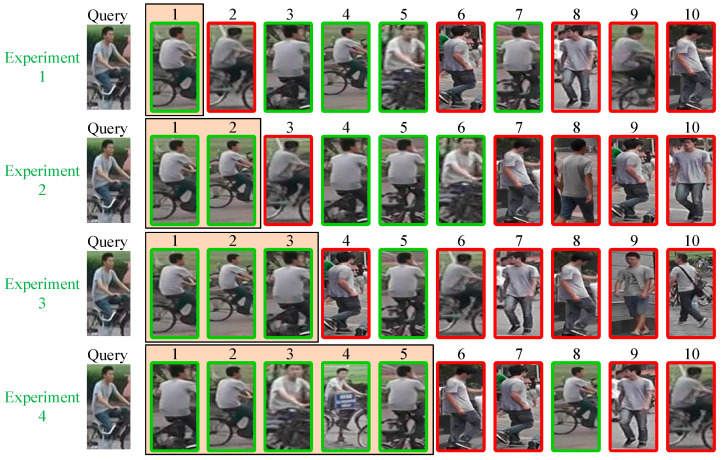
Visualization of ablation experiment results for each component in the training strategy.

**Figure 12 entropy-25-01154-f012:**
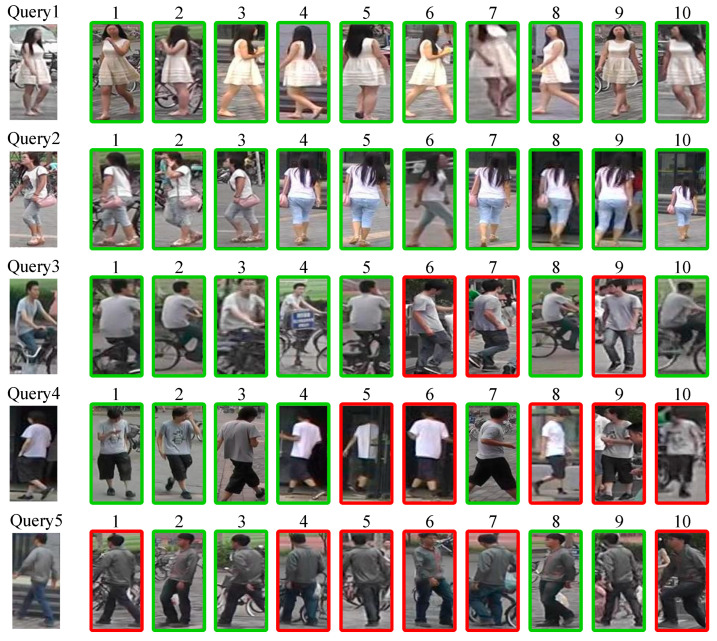
Visualization of person re-identification results using the DDFE network.

**Table 1 entropy-25-01154-t001:** Introduction to the datasets.

Dataset	Number of Cameras	Training Set		Test Set	
		Number of Persons	Number of Images	Number of Persons	Number of Images
Market1501	6	751	12,936	750	19,732
MSMT17	15	1040	32,621	3010	82,161

**Table 2 entropy-25-01154-t002:** Comparison with existing methods.

Methods	Venue	Backbone during the Inference Stage (Params)	Training Cost	Market1501		MSMT17	
				mAP	Rank1	mAP	Rank1
CF-ReID [29]	CVPR2021	ResNet50 (25.5 M)	-	87.7	94.8	-	-
PGANet-152 [30]	ACM MM2021	ResNet152 (60.2 M)	-	89.3	95.4	-	-
CAL [17]	ICCV2021	ResNet101 (44.6 M)	-	89.5	95.5	64	84.2
TransReID [5]	ICCV2021	ViT-Base (86 M)	Nvidia Tesla V100	88.9	95.2	67.4	85.3
TMP [15]	ICCV2021	ResNet101 (44.6 M)	4 × Nvidia Tesla V100	90.3	96	62.7	82.9
DCAL [31]	CVPR2022	ViT-Base (86 M)	Nvidia Tesla V100	87.5	94.7	64	83.1
LTReID [32]	TMM2022	ResNet50 (25.5 M)	4 × GeForce GTX 1080 Ti	89	95.9	58.6	81
UniHCP [33]	CVPR2023	ViT-Base (86 M)	-	90.3	-	67.3	-
CLIP-ReID [34]	AAAI2023	ViT-Base (86 M)	-	90.5	95.4	**75.8**	**89.7**
DC-Former [35]	AAAI2023	ViT-Base (86 M)	4 × Nvidia Tesla V100	90.4	96	69.8	86.2
DDFE	Ours	ConvNeXt v2 Tiny × 2 (57.2 M)	Nvidia Titan V + Nvidia GeForce RTX 2080	**91.6**	**96.1**	69.9	87.5

**Table 3 entropy-25-01154-t003:** Performance comparison: sub-network descriptors vs. integrated feature.

Feature	mAP	Rank1	mINP
$f_{i}^{1}$ (sub-network 1)	91.3	96.1	73.9
$f_{i}^{2}$ (sub-network 2)	91.4	**96.2**	73.9
f (integrated feature)	**91.6**	96.1	**75.1**

**Table 4 entropy-25-01154-t004:** Component ablation experiments of the training strategy.

Number	DropPath	ITM	CurricularFace	mAP	Rank1	mINP
1	-	-	-	89.3	95.2	69.1
2	√	-	-	89.7	95.7	69.7
3	√	√	-	90.4	96	71.3
4	√	√	√	**91.6**	**96.1**	**75.1**

**Table 5 entropy-25-01154-t005:** Generalizability analysis experiments.

Feature Extractor	Year	Parameters	Configuration	mAP	Rank1	mINP
ResNet-50 [36]	2015	25.5 M	Single Network	86.5	94.3	60.6
			Dual Network	**89.2**	**95.7**	**67.8**
MogaNet-T [38]	2022	5.5 M	Single Network	85.9	93.8	61.6
			Dual Network	**86.8**	**94.4**	**62.4**
HorNet-T (GF) [39]	2022	23 M	Single Network	86.9	94.4	63.3
			Dual Network	**88.8**	**95.6**	**66.6**

## Data Availability

The data that support the findings of this study are available online. These datasets were derived from the following public resources: [Market1501, MSMT17].

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
