# Peer review of "Person Re-Identification Method Based on Dual Descriptor Feature Enhancement"

_entropy, 2023, doi:10.3390/e25081154_

Round 1

Reviewer 1 Report

Abstract:

- Specify the limitations of existing person re-identification methods and their impact on recognition performance.

- Describe the specific aspects of human multi-perspective observation abilities that the proposed DDFE network aims to emulate.

- Provide more information about the two sub-networks used in the DDFE network and how they extract descriptors from the same person image, as well as how these descriptors are combined.

- Elaborate on the training strategy employed in the DDFE network, including the CurricularFace loss and the DropPath operation.

- Describe the Integration Training Module (ITM) in more detail and its role in enhancing the discriminability of integrated features.

- Include information about the evaluation metrics used and the achieved state-of-the-art performance on the Market1501, DukeMTMC-ReID, and MSMT17 datasets.

- Mention any limitations or potential drawbacks of the proposed method and future directions for improvement.

Introduction:

- Clearly state the objective of person re-identification and its role in real-time pedestrian tracking and trajectory analysis.

- Provide a comprehensive overview of the current state of research in person re-identification, focusing on deep learning approaches.

- Emphasize the limitations of existing methods in capturing comprehensive and diverse information about a person's appearance and their impact on recognition accuracy.

- Clearly introduce the proposed DDFE network and its key idea of using two sub-networks to extract features from different viewpoints.

- Explain the importance of accurate descriptions from each observer and how they relate to the training strategy.

- Provide details about the CurricularFace loss and the DropPath operation and their roles in improving recognition accuracy.

- Explain the purpose and functionality of the Integration Training Module (ITM) in merging feature descriptors.

Related Work:

- Provide a clear introduction to the two main categories of person re-identification methods: representation learning-based and metric learning-based.

- Elaborate on global feature-based methods and local feature-based methods, explaining their approaches and challenges.

- Explain metric learning-based methods, mentioning the use of cross-entropy loss and triplet loss in pedestrian re-identification.

- Introduce face recognition loss functions and their role in enhancing feature discriminability.

- Connect the discussion of face recognition loss functions to the proposed person re-identification network training.

The sentences are grammatically correct, and the vocabulary is appropriate for the subject matter. Overall, the language used in the paper is of high quality and contributes to the readability and comprehension of the research presented.

Author Response

Response to Reviewer

Dear Reviewer:

    We feel great thanks for your professional review work on our article. Based on the suggestions of other reviewers, we have made some revisions to the article. The changes are as follows:

  1. Due to privacy concerns, all content related to the Duke-MTMC dataset has been removed.

  1. The Abstract section has been modified to improve clarity.

  1. Detailed explanations have been provided for various aspects in the paper, including ConvNeXt V2 Tiny, DropPath, mAP, mINP, Rank-k, and the WarmUp strategy.

  1. Certain images in the paper have been modified to enhance their visual appeal.

  1. In the "4.3 Comparison with existing methods" section, additional information such as model parameters and training consumption has been provided for the SOTA methods, enabling a more comprehensive comparison between the proposed method and the SOTA methods.

  1. A new subsection titled "4.4.4 Parameter analysis" has been added to analyze all the hyper-parameters in this study.

Reviewer 2 Report

This paper introduces an approach called the Dual Descriptor Feature Enhancement (DDFE) network to extract features from an image containing a person and acquired from different viewpoints and generating two descriptors of it. Thus, these features are fused to obtain a feature representation for image matching in person identification tasks. In addition, the network architecture was trained using CurricularFace loss, the DropPath strategy, and the Integration Training Module. Finally, the proposed method was trained using the datasets: Market1501, DukeMTMC-ReID, and MSMT17.

After reviewing your work, I respectfully suggest some modifications to your paper. Following, I list some points to be addressed to improve the manuscript:

1) Please describe detailed the ConvNeXt V2 Tiny architecture.

2) Please describe detailed the DropPath strategy.

3) Please define all the acronyms before use them, for example, BF.

4) Figure 5b is not clear. Can you give more information?

5) Please report the definitions and meaning of the metrics: CMC, mAP, mINP, and Rank-k.

6) Please report how the training parameters were defined for each dataset.  

7) Please give more information about the WarmUp strategy.

8) In Table 2, you reported that your proposal overcomes the selected stat-of-the-art works. What about of the resources used to the trainning? Can you compare the training time and resources used by your proposal against the other ones?

9) Results in Fig. 8 are not conclusive; the number of correct and incorrect recognition is very similar in all rows. Please give more details or include another example.

10) The paper could be improved by providing more detailed information regarding the selected state-of-the-art methods used for comparison.

11) Please define the SOTA performance mentioned in the abstract.

12) The abstract must include some results highlighting the obtained metrics.

Author Response

Response to Reviewer

Dear Reviewer:

        We feel great thanks for your professional review work on our article. As you are concerned, there are several problems that need to be addressed. According to your nice suggestions, we have made extensive corrections to our previous draft, the detailed corrections are listed below.

Point 1: Please describe detailed the ConvNeXt V2 Tiny architecture.

Response 1: We have provided a brief overview of the ConvNeXt V2 Tiny architecture. However, as this is a model developed by others, we did not delve into detailed explanations. If you feel that a more comprehensive description is necessary, please feel free to provide us with further suggestions and we will be happy to accommodate.

Point 2: Please describe detailed the DropPath strategy.

Response 2: We have provided a more detailed explanation of DropPath. If there are any unclear parts in the description, please kindly point them out, and we will continue to make improvements accordingly.

Point 3: Please define all the acronyms before use them, for example, BF.

Response 3: We have made corrections to the abbreviations used in this paper.

Point 4: Figure 5b is not clear. Can you give more information?

Response 4: We have provided a more detailed description of DropPath and have updated the corresponding illustrations. If there are still any unclear points, please kindly indicate them, and we will continue to make further improvements.

Point 5: Please report the definitions and meaning of the metrics: CMC, mAP, mINP, and Rank-k.

Response 5: We have provided more accurate definitions and descriptions for the evaluation metrics, including Rank-k, mAP, and mINP. Rank-k and mAP are widely used metrics in various computer tasks, and we have not provided detailed descriptions for them. However, for the mINP metric, we have supplemented its definition and formula with more detailed information.

Point 6: Please report how the training parameters were defined for each dataset.

Response 6: We have added a new subsection titled "4.4.4 Parameter Analysis" where we conducted experimental analyses on all training parameters of the DDFE network. We have determined the optimal training parameters for each dataset based on the experimental findings.

Point 7: Please give more information about the WarmUp strategy.

Response 7: We have made additional explanations and clarifications regarding the WarmUp strategy. These additions aim to provide readers with a more intuitive understanding of the WarmUp strategy.

Point 8: In Table 2, you reported that your proposal overcomes the selected stat-of-the-art works. What about of the resources used to the trainning? Can you compare the training time and resources used by your proposal against the other ones?

Response 8: Due to the lack of publicly available code for most of the methods listed, it is challenging to directly compare training times. However, we have provided information on the model training parameters during the inference phase and training cost during the training phase. This information was obtained from the original papers, allowing for a partial comparison of the training time and resource utilization between the DDFE network and other existing methods.

Point 9: Results in Fig. 8 are not conclusive; the number of correct and incorrect recognition is very similar in all rows. Please give more details or include another example.

Response 9: We have made some slight modifications to the illustrations to improve their clarity. Additionally, we have provided more detailed descriptions in the text, offering additional context and details. With these enhancements, the experimental presentation should clearly convey the intended content and conclusions.

Point 10: The paper could be improved by providing more detailed information regarding the selected state-of-the-art methods used for comparison.

Response 10: In response to question 8, we have provided the inference phase backbone information and training cost details for the state-of-the-art methods. With these additional details, the descriptions of these SOTA methods should now be quite comprehensive.

Point 11: Please define the SOTA performance mentioned in the abstract.

Response 11: Thank you for your suggestion. We have made revisions to the Abstract section. If you think there are still areas that need improvement, please do not hesitate to point them out, and we will continue to make the necessary refinements.

Point 12: The abstract must include some results highlighting the obtained metrics.

Response 12: As with question 11, we have made modifications to the Abstract section.

Point 13: Other revisions.

Response 13: In addition to the aforementioned modifications, we have removed the content related to the Duke-MTMC dataset from the paper due to privacy concerns.

Round 2

Reviewer 1 Report

  • The revised abstract is clear and provides a good overview of the paper. However, consider helping the reader make a sound understanding of how the following contributions help to achieve the overall objective.

    1. CurricularFace loss to enhance the recognition accuracy
    2. DropPath operation to introduce randomness during sub-network training and promote difference between the descriptors.

  • 3. Integration Training Module (ITM) to enhance the discriminability of the integrated features

    This will help readers understand the novelty and significance of the approach from the beginning.

    Instead of using references [1], [2], [3-5], [6-8], [9], and [10] in the introduction, consider briefly summarizing the key points or insights from these references to support the discussion. This will make the introduction more self-contained and easier to follow for readers.

    The related work section provides a good foundation, but providing more specific examples and discussing the strengths and weaknesses of different approaches will enhance its value for readers.

    In methodology, consider clarifying the purpose of the Weighted Regularization Triplet (WRT) loss and how it contributes to optimizing the inter-class and intra-class distances.

Reviewer 2 Report

Thank you for addressed all my concerns.

I only have a recommendation:

* Change SOTA for State-of-the-art or define the meaning of SOTA in the abstract.